# The Effect of Nanofillers on the Functional Properties of Biopolymer-Based Films: A Review

**DOI:** 10.3390/polym11040675

**Published:** 2019-04-12

**Authors:** Ewelina Jamróz, Piotr Kulawik, Pavel Kopel

**Affiliations:** 1Institute of Chemistry, University of Agriculture in Cracow, Balicka Street 122, PL-30-149 Kraków, Poland; ewelina.jamroz@urk.edu.pl; 2Department of Animal Products Processing, University of Agriculture, Balicka Street 122, PL-30-149 Kraków, Poland; kulawik.piotr@gmail.com; 3Department of Chemistry and Biochemistry, Faculty of AgriSciences, Mendel University in Brno, Zemedelska 1, CZ-613 00 Brno, Czech Republic; 4Central European Institute of Technology, Brno University of Technology, Purkynova 123, CZ-612 00 Brno, Czech Republic

**Keywords:** biopolymer films, nanofillers, functional properties, film mechanical properties, film permeability, antimicrobial activity, nanocomposite materials, food packaging systems

## Abstract

Waste from non-degradable plastics is becoming an increasingly serious problem. Therefore, more and more research focuses on the development of materials with biodegradable properties. Bio-polymers are excellent raw materials for the production of such materials. Bio-based biopolymer films reinforced with nanostructures have become an interesting area of research. Nanocomposite films are a group of materials that mainly consist of bio-based natural (e.g., chitosan, starch) and synthetic (e.g., poly(lactic acid)) polymers and nanofillers (clay, organic, inorganic, or carbon nanostructures), with different properties. The interaction between environmentally friendly biopolymers and nanofillers leads to the improved functionality of nanocomposite materials. Depending on the properties of nanofillers, new or improved properties of nanocomposites can be obtained such as: barrier properties, improved mechanical strength, antimicrobial, and antioxidant properties or thermal stability. This review compiles information about biopolymers used as the matrix for the films with nanofillers as the active agents. Particular emphasis has been placed on the influence of nanofillers on functional properties of biopolymer films and their possible use within the food industry and food packaging systems. The possible applications of those nanocomposite films within other industries (medicine, drug and chemical industry, tissue engineering) is also briefly summarized.

## 1. Introduction

Nanocomposite films are a new generation of packaging materials with combination of bio-based polymer and fillers that have at least one nanometer scale dimension. In nanocomposite films, the biopolymer acts as a matrix, while the nanofillers are dispersed therein to improve the functional properties. Nanocomposites have a set of improved properties, such as mechanical, antimicrobial, or physical properties. These properties do not occur naturally in the biopolymers themselves, therefore they are gained due to addition of nanocomposites [1]. The components of biopolymer films are characterized by high availability and good biodegradability. However, one-component films have relatively poor mechanical properties and high water vapor permeability, which may cause them not to be of sufficient quality to be used as packaging materials. This can be reduced by mixing two biopolymers with one another [2]. Currently, a novel method to improve the properties of biopolymer films is the use of nanofillers, which can fulfill not only the reinforcing function but also could act as an active ingredient. In recent years, the concept of active agents for biopolymer films has received much more attention. Such active ingredients in biopolymer films can extend the shelf-life of food products, through exhibiting antimicrobial and/or antioxidant activities [3,4]. The preservative effect of biopolymer films with nanofillers have been reported for the whole spectrum of food products including vegetables and fruits, mushrooms, dairy products, meat and meat products, and fish and other seafood [5,6,7,8,9,10,11,12,13,14,15]. The development of nanotechnology has led to the design of nanocomposite film materials in which nanofillers play an active role. The aim of this review is to summarize recent advances and achievements regarding the addition of nanofillers—such as clay, metals, metal oxides, polymer nanoparticles, nanocellulose, etc.—into biopolymer films with a focus on their functional properties and a possible use in food packaging systems. Additionally, this review shows recent applications of biopolymer films with nanofillers on wound dressing, drug, and enzyme delivery systems and tissue engineering.

## 2. Types of Biopolymers and Nanofillers

New trends and perspectives in nanotechnology, have facilitated the path to use nanofillers as active agents in the packaging industry. This solution applies to the production of biodegradable packaging material, which is based on biopolymers and nanofillers. This type of nanocomposite material is not capable of replacing synthetic packaging materials, because it has many disadvantages, i.e., weak mechanical properties, too high hydrophilicity, and susceptibility to decomposition. However, they are interesting alternatives to plastic materials and can be used for example in areas where plastic recovery is not economically feasible [16].

Currently, there is a growing interest in the combination of biopolymer and nanofiller-based materials. Nanofillers can have different shapes and sizes, but their individual particles size is, by definition of the nanomaterials, below 100 nm [1]. The size of nanofillers is very beneficial for nanocomposite materials because they are based on a large surface area which leads to a large interphase or boundary area between the biopolymer matrix and nanofiller. Due to such interaction, the biopolymer matrix is modified, which could contribute to the improvement of mechanical, thermal, and barrier properties of bionanocomposite materials [17].

### 2.1. Types of Biopolymers Matrix

Bio-based polymers, which are considered the matrix base during preparation of nanocomposite films, have many advantages. They are renewable, biodegradable, multi-functional, and biocompatible [18].

Bio-based polymers can be classified into two major categories:Natural bio-based polymers, including:
-polymers extracted from agricultural resources:
■polysaccharides:
neutral: e.g., cellulose, hemicellulose, starch cationic: e.g., chitin, chitosananionic: alginic acid, hyaluronic acidof bacterial origin: e.g., pullulan, carrageenan; ■proteins: e.g., gelatin and whey protein; Other bio-based polymers: e.g., lipid, lignin, natural rubber, urushiol, DNA, etc. polymers produced directly from microorganisms bacterial cellulose; polyhydroxyalkanoates, poly-ɛ-caprolactones. Synthetic bio-based polymers, including:
■natural-based or bio-based synthetic polymers, the monomers of which are derived from renewable resources (e.g., poly(lactic acid)-PLA);■partially bio-based polymers such as polyethylene (PE), poly(ethylene terephtalate) (PET) and polyamide (PA), etc. [19,20,21,22].

Among biopolymers, starch and chitosan are the most often used polysaccharides in the production of biopolymer films. Chitosan is a highly interesting film forming base due to its biological (antimicrobial and antioxidant activity) and physical (thermal or mechanical) properties [23]. The use of starch as film forming coatings eliminates the problem of environmental pollution because it is an economical, non-toxic, and environmentally friendly biopolymer. However, starch’s poor mechanical properties, moisture sensitivity, and weak barrier properties restricts its use as a commercial packaging materials [24]. Pullulan is a neutral polysaccharide obtained from the fermentation medium of the fungus-like yeast *Aureobasidium pullulans* [25]. It has superb optical properties, and applications of this type of non-toxic, biodegradable polymer for use in novel optical materials may be encouraging [26].

Proteins such as gelatin are also used in the preparation of films. However, proteins have poor mechanical properties and very high sensitivity to moisture which are problems in the commercialization of this type of film [27]. Beeswax is a good barrier against moisture migration, because it is hydrophobic and has a firmly packed crystalline structure, which is why it is often added to a polysaccharide and protein matrix to reduce the parameters of water vapor permeability of biopolymer films [28,29]. 

Poly(lactic acid) or polylactide is derived from renewable biomass products and wastes such as corn starch. This polymer is compostable, non-toxic, biocompatible, thermoplastic, and has desirable mechanical properties [30]. However, PLA has some limitations for packaging applications such as weak water vapor permeability, low thermal stability, and high rigidity [31]. Polyvinyl alcohol (PVA) is a non-toxic polymer, which has excellent chemical resistance and physical properties [32]. The major drawback of this synthetic polymer is its low mechanical strength [33]. Polyhydroxyalkanoates (PHAs) are a family of biopolyesters produced by a wide variety of bacteria, which are often used as a film component [34]. Due to its biodegradability and compatibility with various polymers and nanofillers, poly(ɛ-caprolactone) (PCL) has recently gained interest [35]. However, due to the hydrophobicity and crystallinity of this polymer, PCL undergoes very slow biodegradation as a result microorganisms in the environment [36]. 

As noted, there are many types of biopolymer films with each having their own advantages and disadvantages (Table 1). Some drawbacks of the polysaccharide or protein films can be overcome. Blending the different biopolymers together is one of the promising methods of improving the functional properties of biopolymer-based films [37,38]. The second is to use cross-linking of biopolymer films with various nanofillers.

### 2.2. Types of Nanofillers

There are four types of nanofillers: clays, organic, inorganic, and carbon nanostructure. The organic nanofillers include natural biopolymers (e.g. chitosan, cellulose), whereas inorganic agents are either a metal (e.g. silver) or metal oxide (e.g. ZnO and TiO_2_) [4,67]. Carbon nanostructures can be classified into fullerenes, graphene, carbon nanotubes, and nanofibers [68]. A list of examples of properties of various nanofillers is shown in Table 2. 

The hybrid materials, which are materials made from hybrid of organic and inorganic materials, attracts more and more attention in various fields of research. In comparison to conventional materials, organic–inorganic hybrid materials have features derived from the inorganic part (rigidity, dimensional stability and thermal stability) as well as the properties of organic materials (tenacity and workability) [94].

#### 2.2.1. Clay and Organic Nanofillers

Natural clays are inexpensive materials and exist as aluminum silicate, which consists of fine-grained minerals. Currently, nanoclays are the most commonly used nanoparticles, and they exist in the form of sheets/platelets, which have at least one dimension in the nanoscale range. The most often used nanoclay is montmorillonite (MMT), which consists of two tetrahedral silica sheets connected to an edge-divided, eight-sided aluminum oxide sheet [16]. Another type of nanoclay is halloysite (Hal), which is a natural aluminosilicate clay mineral with hollow, cylindrical-shaped nanotubes [95]. The strong mechanical properties of Hal have been integrated with the antimicrobial and photocatalytic properties of metal oxides and metal nanoparticles by grafting them on Hal [95,96,97]. 

Natural biopolymer nanofibrils, which are composed of various biopolymer molecules—i.e., cellulose, collagen, and chitin—are often used due to their biocompatibility, biodegradability, durability, availability, and unique mechanical properties [98]. Nanocellulose, which is extracted from cellulose, is characterized by the reactive surface of hydroxyl groups and thus can be adapted to different surface properties [99,100]. There are three types of cellulose used on the nanoscale as an additive to biopolymer films: nanocrystalline cellulose, nanofibrillated cellulose, and bacterial nanocellulose [100]. These types of nanocellulose differ in their morphology, degree of crystallinity, particle size, and some other properties. These differences result from their sources and various extraction methods used [100]. Nanocrystalline cellulose with its high crystallinity and short-rod shape is also known as cellulose nanocrystals or cellulose nanowhiskers and is usually extracted from cellulose fibrils by acid hydrolysis [101]. The source of cellulose nanocrystals may also affect the functionality of the nanofiller [102]. 

Nanofibrillated cellulose is extracted from cellulose fibers using mechanical methods. This long, flexible, and tangled nanocellulose is also known as cellulose microfibril, microfibrillated cellulose, cellulose nanofiber, cellulose nanofibril, and nanofibrillar cellulose [100].

Bacterial nanocellulose (BNC), which is mainly extracted from cultures of the Gram-negative bacteria, *Gluconacetobacter xylinus*, has a higher molecular weight and crystallinity than cellulose from plant sources [103]. The different types of nanocellulose have been incorporated to many biopolymer films [104,105,106,107].

Chitosan and chitin nanoparticles obtained respectively from chitosan or chitin have gained attention as nanofillers, due to their attractive surface area, biocompatibility, non-toxicity and film forming ability [108,109,110,111,112]. An important limitation in the use of chitosan nanoparticles is their poor stability. It can be improved by controlling the conditions e.g., by changing the structure with chemical agents. Poor solubility of chitosan nanoparticles is the next limitation, which is a major problem in the encapsulation of hydrophobic drugs [113].

#### 2.2.2. Inorganic Nanofillers

Due to their better mechanical, thermal, physical, biological, and chemical properties than bulk materials, nanoparticles (NPs) are an interesting solution as functional agents for biopolymers films. Copper nanoparticles (CuNPs) are known to possess antimicrobial activity on wide spectrum of microbes. Released copper ions have a high redox potential and the ability to destroy and cause apoptosis of microbial cell components [114]. Owing to their excellent physicochemical and biological properties, silver nanoparticles (AgNPs) have gained increasingly more attention for use in wound healing [115], food packaging [116], and in medical applications [117]. Selenium nanoparticles (SeNPs) show significantly reduced toxicity compared to selenium. In addition, SeNPs can be used as a platform for transporting different drugs to target destinations [118].

Metal oxide nanoparticles such as zinc oxide (ZnO NPs), titanium oxide (TiO_2_ NPs), silica (SiO_2_ NPs), aluminum oxide (Al_2_O_3_ NPs), cerium oxide (CeO_2_ NPs), iron oxide (Fe_2_O_3_ NPs), and copper oxide (CuO NPs) have been added to biopolymer films as active nanofillers. TiO_2_ NPs and ZnO NPs have photocatalytic antibacterial properties, caused by reactive oxygen species (ROS) formation after exposure to UV-light [119]. Additionally, ZnO NPs can release zinc ions that damage bacterial cells, showing that antimicrobial properties do not have to be dependent on UV exposure [120]. The low hydrophilicity of TiO_2_ prevents the penetration of moisture into the biopolymer matrix. Moreover, the barrier effect of TiO_2_ leads to reduction in water vapor permeability [121]. CuO NPs have antimicrobial, antibacterial, and antioxidant activity and exhibit a UV-blocking effect [122,123]. Tin oxide (SnO_2_) nanoparticles have electrical, thermal, mechanical and gas barrier properties [124,125]. The addition of non-toxic and neutral Al_2_O_3_ NPs to the biopolymer matrix provides a wide range of promising properties due to its small particles, significant surface area, and good activity [126,127].

#### 2.2.3. Carbon Nanofillers

Graphene-based materials have gained attention due to their properties which include excellent mechanical properties and substantial electron mobility. Graphene, in comparison to other carbon-based nanomaterials (for example carbon nanotubes), is characterized by a larger surface area that could facilitate interactions with the polymer matrix [128,129]. Graphene oxide (GO) is the most promising nanofiller, among the graphene-family nanomaterials, because it has a lower tendency to agglomerate than pristine graphene [130]. The biggest limitations, for the use of GO, are its intrinsic zero band-gap energy and low solubility in organic and aqueous solvents [131]. The chemical functionalization of graphene oxide leads to its modification and thus increases the possibilities of its potential application. Reduced GO (RGO) has very good mechanical, physical, electrical, and thermal properties, which makes it attractive nanofiller in biopolymer films [132]. 

#### 2.2.4. Other Nanofillers

Semiconductor quantum dots (QDs) are attracting increasing attention due to their optical properties and high photostability [133]. Due to their strong tendency of oxidation and agglomeration, it is desirable to improve the compatibility and stability of this type of nanofillers in biopolymers [134]. Graphene quantum dots (GQDs) incorporated in biopolymer films have advantages such as availability, biodegradability, low price, and low production cost, making the nanocomposite films a preferred candidate for use in optoelectronics applications [135].

## 3. Effects of Nanofillers on the Functional Properties of Biopolymer-Based Films

The graphical illustration of nanofillers use within the biopolymer matrix along with their functional properties is presented in Figure 1. 

### 3.1. Effects of Nanofillers on the Physical Properties of Polymer-Based Films

Water resistance is an important functional parameter of biopolymer films. In many studies, water resistance of the films is evaluated by their water solubility, swelling degree, water content, and water vapor permeability [87,136]. 

The addition of nanofillers decreases the solubility of biopolymer films due to the ratio of dimensions and crystalline areas of the fillers [87,137]. Noshirvani and co-workers (2018) confirmed that the lower solubility of the nanocomposite films is caused by the hydrogen bond between the hydroxyl groups of starch, PVA, and cellulose nanocrystals. The three-dimensional network generated leads to the strengthening of the network and reduces its solubility [138]. The same trend was observed in kefiran-caboxymethyl cellulose films with CuO NPs [139] and in kefiran films with Al_2_O_3_ NPs [127]. Moreover, the type of nanoparticles also plays an important role in the solubility of the film. A decrease in the solubility of the film can be attributed to the very low solubility of the nanoparticles compared to the polymer chains, which leads to a reduction in the hydrophilicity of the biopolymer matrix [121]. The interactions between halloysite nanoclay and soluble soybean polysaccharide caused a decrease in water solubility due to reduction in the availability of hydroxyl groups to interact with water [140].

The empty spaces in the structural network of nanocomposites can be occupied by water molecules. This phenomenon is considered the water content of the film [87,141]. The addition of nanofillers into the films can cause a decrease in water content. This behavior could be related to the interaction between nanofillers and functional groups of the biopolymer chain, which can lead to a reduction in the available spaces in the biopolymer matrix [87]. Noshirvani and co-workers (2018) developed starch-PVA films with cellulose nanocrystals (CNC). They noticed that the addition of the nanofiller caused a decrease in the moisture content of the film. Such behavior was demonstrated by the crystal structure of the CNC, which causes lower water uptake than the polymer matrix and the formation of strong hydrogen interactions between the CNC and the starch-PVA matrix [138]. The addition of nano-SiO_2_ into whey protein isolate–pullulan matrix decreased moisture content of nanocomposite films. The strong interactions between components caused the diffusion of water molecules to nanocomposites [142].

An important property of biopolymer films, and in particular polysaccharide films, is the swelling degree, which determines the amount of absorbed water. As the value of the swelling degree increases, the tolerance of film to water increases [143]. In general, the addition of nanofillers improves the swelling degree and increases the water resistance of the films. This is attributed to the strong hydrogen interactions between the nanofiller/nanofiller and the nanofillers/biopolymer matrix [143]. However, the reverse trend can also be noticed. The addition of chitin nanofiber (CHNF) caused an incremental swelling degree of chitosan films, which was due to increased amount of OH^-^ groups from CHNF and resulted in increased water absorption [144]. The same phenomenon was observed in the chitosan/starch films with halloysite nanotubes, which was attributed to the increased porosity and hydrophilicity of the nanocomposite films [145].

The water vapor permeability (WVP) is very important for fresh food products and for products where dehydration and absorption of moisture should be avoided. The values of WVP of packaging systems should be at the lowest possible level [28,87,146]. The water vapor permeability in biopolymer films is influenced by the chemical nature of macromolecules, porosity and crystallinity, degree of cross-linking, relative humidity, and addition of a plasticizer [147,148]. The addition of nanofillers to biopolymer films has an impact on the water vapor barrier properties. The low WVP of films is an important feature in the packaging of food products as they can reduce the moisture transfer between the inner and outer packaging environment [149]. The presence of impermeable nanoparticles in the biopolymer matrix prevents the mobility of the biopolymer chain, and consequently can lead to a reduction in the water vapor permeability of the nanocomposite films [28]. In addition, the hydrophobic character of nanoparticles (e.g., TiO_2_) affects the reduction of the WVP of the film, due to the low aspect ratio of this nanoparticle and the irregular strengthening of the chains [87]. Shankar et al. concluded that the presence of halloysite nanotubes and AgNPs creates an increased tortuous path for the passage of water vapor molecules through the alginate matrix. On the other hand, exceeding the critical concentration of nanofillers increases the water vapor permeability of nanocomposite films [150]. Similar patterns of changes in WVP properties have been observed in many studies [87,95,143,151,152,153].

The type of nanofiller used also has an effect on the WVP of films. Zahedi and co-authors (2018) established that sodium MMT nanoclay is more effective in the reduction of the WVP of carboxylmethyl cellulose films than ZnONPs. These differences may result from various structural and arrangement features. ZnONPs have a hexagonal close packaged structure, while MMT is a layered silicate structure [154]. Nanoclays impede the diffusion of the water vapor due to impermeable layers of this type of nanofiller [155,156]. However, the addition of another type of nanoclay, laponite—synthetic hectorite-like clay—to kafirin film, does not significantly affect the WVP of the tested films. The hydrophilic clay, due to the presence of Si-OH groups, can affect the hydrophilicity of the surface [157]. The same trend was observed in collagen films with laponite [158].

In related literature, the opposite effect on the water barrier properties of biopolymer films is also reported. The addition of MMT/alkylammonium (hexa- and tetra-decyltrimethylammonium) and MMT/chitosan to cellulose acetate coatings increased the WVP value of the film [159]. Also the capping agent used in the preparation of the nanoparticles is critical for the WVP of the film. CMC was used as a capping agent in preparation of ZnO NPs. The nanoparticles caused an increase in the WVP of the gelatin films, which is attributed to the hydrophilic nature of CMC used as capping agent [160]. Silva and co-workers (2019) prepared new nanocomposites based on laponite and cellulose nanofibers. They concluded that the increase in WVP is related to the hydrophilic nature of laponite [161].

Oxygen permeability (OP) is one of the crucial parameters of films used in food packaging. Low OP values are preferred, because oxygen can cause deterioration in the quality of packed food products [162]. The addition of nanofillers into biopolymer films may cause a decrease in the OP values. Wu et al. (2019) stated that the AgNPs addition into nanocellulose films with grape seed extracts caused the improvement in barrier properties of the tested films. AgNPs filled the interspaces of matrix and hindered the transfer of O_2_ molecules through the film [163]. The presence of nanocellulose also causes the reduction of OP of the film, which can be attributed to the formation of a dense network structure of the film matrix [152]. A similar phenomenon was observed with the addition of halloysite nanoclay into potato starch films. The reduction in the permeability values may result from the fact that gases have an extended diffusion path [164]. 

### 3.2. Effects of Nanofillers on the Mechanical Properties of Polymer-Based Films

Two main parameters which are often used to determine the mechanical properties of biopolymer films are tensile strength (TS, MPa), which is used to measure strength, and the percentage of elongation at break (EAB, %) which is used to determine the elasticity of the film. These parameters must meet certain standards to maintain integrity during packaging [149]. The addition of nanoparticles to biopolymer films significantly affects their mechanical properties. This may be due to the fact that nanoparticles have a very large specific surface area that can affect interfacial strength and degree of dispersion. Even distribution of nanoparticles within the biopolymer matrix results in a specific transfer of stress through the shear mechanism from the biopolymer matrix to the nanoparticles and could result in effective load transfer and increased strength of the film [31,165].

The phenomenon of strengthening the tensile strength may also result from the interaction between the nanofiller and the biopolymer matrix, where hydrogen and covalent bonds between nanoparticles and, for example, hydroxyl groups of biopolymer are formed, which leads to the strengthening of molecular forces between nanoparticles and the biopolymer [143,166].

Nanofillers can fill the free space between biopolymer chains, which increases the intermolecular attraction force, making the biopolymer matrix very dense and less permeable [143]. Improvement in TS of polymer films by addition of nanofillers has been observed in films from konjac glucomannan [167], fenugreek seed gum [168], alginate [95], chitosan [169], agar [150], starch [121,143], soy protein [152], whey protein [87], and CMC [170].

Reinforcement of biopolymer films with different nanofillers have been studied by others. In the presence of nanoclay in a gelatin matrix, the values of TS of gelatin films improved [171]. The same trend was observed in starch-polyvinyl alcohol films with cellulose nanocrystals (CNC) [138]. The authors observed improvement of mechanical properties of these films due to the stiffness of the CNC and the formation of hydrogen bonds between the CNC and the polymer matrix. Popescu and co-workers found that incorporating rigid molecules of CNC into starch-PVA films induces orientation of the polymeric molecules and favors the formation of H-bonds between the polymeric chains [172]. Oun and Rhim (2016) isolated cellulose nanocrystals from rice, wheat, and barley straws. They concluded that properties of carboxymethyl cellulose (CMC) films varied depending on the type of cellulose nanocrystals. The addition of rice and barley straw into CMC film improved tensile strength parameters. However, the best results of WVP were achieved in CMC films with wheat straw [102]. Also, the combination of different nanofillers affects the mechanical properties of films. The binary films (PLA-AgNPs and PLA-CNC) showed lower values of mechanical properties than ternary films (PLA/AgNPs/CNC). The addition of AgNPs and CNC into PLA films improves their tensile strength and modulus values, which is due to synergistic effect between AgNPs and CNC [107].

The changes in elongation at break (EAB) exhibit the opposite trend than TS, when the nanofillers are added into the film polymer matrix. Since the addition of nanoparticles causes a decrease in the EAB value. Therefore, the addition of nanoparticles can increase the strength of the film but do not affect its flexibility [166]. A similar trend was observed in soy protein films with nanocellulose. The reinforcement may affect the limited movement of the polymer chain by reducing the free volume in the polymer matrix [152].

The critical concentration of nanofillers, above which the TS values of films decrease, should also be mentioned. Too high concentration of NPs causes the biopolymer matrix to be unable to evenly distribute the NPs, resulting in the reduction of strength [173]. The same problem can be observed with the addition of other types of nanofillers such as nanocrystals, which in too high concentration tend to grow together into larger crystals [105] or nanoclays which due to their poor dispersion and high surface energy can cause a decrease in the TS [168]. The decrease in TS at high nanofiller concentrations may also be related to crosslinking decrease in blend film and the damage to the polymer network layers due to the presence of large NP agglomerates [87].

When nanofillers are added to the films, they can cause lower resistance of a polymer matrix and its fracture. This can lead to the formation of micropores in the matrix, change of the crack growth of the path and transformation of the matrix shape. The formation of small clusters of nanoparticles increases the number of cracks and contributes to creating holes with a role in the energy loss and flexibility mechanism. The reduction of the polymer–nanoparticle interface surface may be due to the increase in the size of the nanoparticles. Thus, concentration at higher levels of nanoparticles leads to an optimal level of plasticity and reduced elasticity [87].

The increase in the TS value and the decrease in the EAB value may also be related to the moisture content of the tested biopolymer films. Water can act as a plasticizer in films and a decrease in plasticizer content reduces the elasticity of the film, which may lead to an increase in TS and Young modulus (YM) and a reduction in EAB [171,174].

Vicentini and co-workers (2018) studied the effect of modified carbon nanostructures (p-methoxyphenyl functionalized, carbon nanohorns, and reduced graphene oxide) in poly(L-lactic acid) composites. They concluded that the mechanical properties depended on the type of nanofiller used, and these differences can be attributed to the different size and shape of nanofillers [175].

The presence of graphene oxide in the cellulose carbamate matrix caused improvement in tensile strength of nanocomposite films. The roughness of the larger surface of graphene oxide sheets ensures better adhesion with cellulose carbamate, which could lead to stronger mechanical blocking [129]. A similar enhancement in TS was reported in chitosan/graphene oxide nanosheet films [176]. Son et al. (2015) compared mechanical properties of sodium carboxymethyl cellulose/reduced graphene oxide and sodium carboxymethyl cellulose/graphene oxide films. They concluded that reduced graphene oxide could interact more effectively with sodium carboxymethyl cellulose than graphene oxide because reduced graphene oxide reacts with sodium carboxymethyl cellulose through strong van der Waals interactions and hydrogen bonds due to the presence of oxygen-containing functional groups on the edge of the reduced graphene oxide sheet [177].

The type of biopolymer also affects the mechanical properties (modulus, strength, and toughness) of nanocomposite films. Achaby et al. (2018) studied the effect cellulose nanocrystals addition on the mechanical properties of alginate, chitosan, and κ-carrageenan films. The increase in mechanical strength of the alginate films with the addition of CNC was caused by strong hydrogen and ionic interactions between the free hydroxyl groups of the CNC and the hydroxyl and carboxyl groups of the alginate. The same trend was found in κ-carrageenan films with CNC, which the authors explained as a homogeneous CNC dispersion in the carrageenan matrix due to hydrogen bonds, which creates interfacial interaction between the CNC and the κ-carrageenan [178]. Moreover, Kassab and co-workers (2019) concluded that the increase in the mechanical properties of carrageenan films is related to the specific properties of CNC which was extracted from sugarcane bagasse. The authors explained that the morphology (needle shape), surface functionality, large aspect ratio, and high degree of crystallinity of CNC are responsible for improving the mechanical properties of the κ-carrageenan film [179]. 

Different conclusions were observed in chitosan nanocomposite films. The addition of up to 5 wt % of CNC caused an increase of mechanical properties parameters, which can be attributed to the interaction between the anionic sulphate groups of the CNC and the cationic amide groups of the chitosan. In addition, the nano-reinforcing effect of CNC was achieved through effective stress transfer of the CNC-chitosan interface. The addition of CNC at the highest level (8 wt %) caused a decrease in mechanical strength, which was related to the agglomeration phenomenon of CNC in the chitosan matrix. The addition of CNC also affects the elongation at break. The reduction of the EAB of each type of film is due to the rigid behavior of the CNC and its interactions with biopolymers that lead to a reduction in macromolecular biopolymer chains [178]. Kanmani and Rhim obtained three types of nanocomposite films based on agar, carrageenan, and CMC with ZnONPs. The addition of ZnONPs to the biopolymer films resulted in a decrease in TS and YM. In contrast, ZnONPs addition significantly increased the EAB of biopolymer films. The presence of ZnONPs did not interrupt the movement of polymer chains [180].

### 3.3. Effects of Nanofillers on the Antimicrobial Activity of Polymer-Based Films 

The antimicrobial packaging system is intended to ensure the safety and prolong storage stability of food by preventing the development of microorganisms. The packaging materials can acquire antibacterial activity through:(1)the addition of antibacterial substances to the biopolymer matrix;(2)use of inherently antimicrobial polymer (for example polymer resins or chitosan);(3)irradiation of a biopolymer matrix which results in the formation of reactive oxygen species (ROS) [181,182].

The antibacterial agents can be divided into two types: bactericidal, which kill bacteria; and bacteriostatic, which cause the inhibition of bacteria growth. The incorporation of nanoparticles into biopolymer films increased their antimicrobial activity. The mechanism of antimicrobial activity of nanoparticles depends on the type of nanoparticles as well as on the species and the natural properties of the bacteria. The bacteria’s sensitivity to NPs is influenced by the bacterial cell wall structure and their growth rate. Fast-growing bacteria are more sensitive to NPs than slow-growing bacteria [87,183]. Moreover, the antibacterial activity of nanoparticles is greater in Gram-negative bacteria than in Gram-positive bacteria, which is due to the bacterial cell structure. Gram-positive bacteria have a thick three-dimensional layer of peptidoglycan (~20–80 nm) that creates a specific barrier to nanoparticles and prevents them from entering inside the cell. A thin layer of peptidoglycan (~7–8 nm) in Gram-positive bacteria is an ineffective barrier against released nanoparticles ions [184]. There are several hypotheses of the mechanism of antibacterial action of nanoparticles. The first of these concerns the interaction of negatively charged nanoparticles with biologically active molecules that are found on the surface of cells, which in turn can lead to the leakage of cellular components [85]. The second type of mechanism concerns the interactions of nanoparticles with bacterial DNA, resulting in cell death. The potential mechanisms of antimicrobial activities of nanofillers are illustrated by the example of nanoparticles shown in Figure 2. 

In addition, each type of nanoparticles could work differently. In the presence of ZnONPs, H_2_O_2_ is generated [185], whereas sulfur ions, which are generated from SNPs, cause the formation of toxic H_2_S, which interacts with the -SH group and generates stress ROS [85,186]. The type of nanoparticles also affects their bactericidal activity. Small-sized nanoparticles have a stronger bactericidal effect [187], while a positive surface charge of metallic nanoparticles results in easier interaction with a negative charge of bacterial surfaces [188]. 

In related literature, many antibacterial aspects of nanostructures have been reported. The influence of various types of nanoclays (Cloisite Na+, Cloisite 20A and Cloisite 30B) on the antibacterial properties of whey protein isolate (WPI) films was investigated. Only the presence of Cloisite 30B showed a bacteriostatic effect on *L. monocytogenes*, while none of the different types of nanoclays had any antibacterial effect against *E. coli* [189]. The antibacterial effect of Cloisite 30B can be attributed to the presence of a quaternary ammonium group in the silicate layer, which destroys bacterial cell membranes and causes cell lysis. The same behavior was observed in PLA films with Cloisite 30B [190]. Although hydrophilic NaMMT or organically modified montmorillonite (OrgMMT) do not show any particular antibacterial action, it has been shown that their addition to the film enhances the antibacterial effect of chitosan-PVA films [191].

Chitin nanofibrils (CNF) could be characterized as bacteriostatic rather than as a bactericidal agent. The addition of CNF into carrageenan matrix improved antimicrobial activity of nanocomposite films. The proposed mechanism of CNF activity consists of blocking access to nutrients and oxygen, which causes the bacteria to flocculate [192]. Salari and coworkers (2018) developed active chitosan films with bacterial cellulose nanocrystals (BCNC) and silver nanoparticles AgNPs. The antibacterial and antifungal properties of this type of film were related to the presence of AgNPs, while BCNC and chitosan had no antibacterial or antifungal activity [184]. The mechanism of antibacterial behavior of ZnO nanorods could have a physical basis. Nanorods could act like needles that easily pass through the bacterial cell wall [185]. The same results were reported in sago starch films with ZnO nanorods [193]. 

The high antimicrobial activity of GO is due to the physical destruction of cell membranes due to exceptionally sharp edges of GO and chemical oxidation through generation of ROS. In comparison to other carbon nanomaterials, graphene oxide has low cost of production and causes mild cytotoxicity to mammalian cells in low doses [194,195].

Increased attention is being paid to hybrid materials that give new properties to nanocomposite materials. Hybridization of metal nanoparticles and polymer can improve the properties of individual components. The addition of the hybrid of ZnO NPs-chitosan to starch films resulted in an increase in antibacterial activity against *E. coli* and *S. aureus*, compared to starch films alone [196]. Moreover, when a hybrid of PVA/GO/starch AgNPs was used in PVA matrix, an increase in the antibacterial properties have been observed, which might be due to the synergistic effects of GO and AgNPs in PVA matrix [197]. The addition of GO and CNC stimulates the antibacterial activity of PLA film, resulting in greater activity than single nanofillers [198].

Table 3 presents some recent studies on the impact of nanofillers on functional properties of nanocomposite films.

## 4. Functional Application of Biopolymer-Based Films with Nanofillers

Although many countries have made legal regulations regarding the contact of nanomaterials with food or with human skin, those regulations vary from country to country. Moreover, the definition of ’nanomaterial’ is not even unified between countries. The detailed review of the legal status regulations for nanomaterials in EU has been recently performed by Rauscher et al. [253]. Nanomaterials that are allowed for use in contact with humans are required to have a detailed risk assessment [254]. This chapter is devoted to nanocomposite materials and their use in various areas of life.

### 4.1. Food Preservation Application

Packaging materials made from natural biopolymers are characterized by poor mechanical properties and high WVP parameters. The utilization of nanotechnology in this field may help to improve these parameters as well as give them completely new active properties. The main task of the nanocomposite packaging is to increase the shelf-life of food during storage and distribution [67].

Chitosan–gelatin films with the addition of AgNPs have been used to protect red grapes. Preliminary studies confirmed the active nature of such coatings, extending the storage time of grapes. The addition of 0.05% AgNPs in the chitosan–gelatin films resulted in the extension of the red grapes shelf-life with no signs of molds for 14 days, whereas the concentration of 0.1% AgNPs prolonged the shelf-life to 18 days [200]. In another case, AgNPs were incorporated into chitosan films and prolonged the storage time of litchi from 4 to 7 days [5]. The incorporation of MMT into starch–cashew tree gum increased protection against moisture loss of cashew nut kernels. However, the MMT additive did not achieve the critical value of peroxides for acceptance (10 or 20–30 mEq O_2_ kg^−1^) [6].

Soft white cheese was packed into chitosan-PVA films with TiO_2_ NPs and stored at 7 °C for 30 days. After 15 days of storage, the number of coliforms significantly decreased in the samples coated with films with 2%, 4%, and 8% TiO_2_ (by 1.47, 1.47, 1.30 log cfu/g cheese, respectively, compared to the control (1.90 log cfu/g). The results showed that bionanocomposite films with strong antimicrobial activity against gram positive (*S. aureus*), gram negative (*P. aeruginosa, E. coli*) bacteria and fungi (*C. albicans*) could be used as an environmentally friendly material for food packaging. Additionally, the lack of migration of TiO_2_ from the film to cheese was confirmed, which proves the high safety of this type of material [7]. Nanostructured chitosan–manolaurin films caused a reduction of *L. monocytogenes* population (by 2.3–2.4 log) on ultrafiltered cheese after 14 days [8]. 

The addition of ZnO nanoparticles into mahua oil-based polyurethane/chitosan film significantly influenced antibacterial properties when these films were used for packing carrot slices. After 9 days of storage, carrot slices packed in mahua oil-based polyurethane/chitosan+ ZnO NPs films had a lower degree of bacteria growth (by approx. 0.3–0.6 log cfu/g) than pieces of carrot slices without film and with PE film [213]. Antimicrobial activity against *E. coli* and *L. monocytogenes* was recorded for cooked minced fish paste packaged in PLA films with ZnO NPs and after 10 days of storage, the number of colonies of both bacteria was reduced to zero. The antimicrobial activity might have resulted from direct contact of microorganisms with ZnO NPs or Zn^2+^ ions emitted from the film [31]. The addition of ZnO NPs into chitosan/CMC also positive influenced packaged Egyptian white soft cheese (rheological properties, color measurements, moisture, pH, and titratable acidity) during storage [211]. The starch–halloysite–nisin nanocomposite films effectively protected Minas Frescal cheese against post-process contamination with *L. monocytogenes* [236]. Echeverria and co-workers (2018) investigated the effect of MMT and clove essential oil in starch films on storage of muscle fillets of bluefin tuna (*Thunnus thynnus*). They noticed that the presence of clay prolonged the antimicrobial and antioxidant effects of clove essential oil, while no migration of clay’s metal (Si and Al) into the muscle of the fish were observed [9]. Mathew and co-workers (2019) developed PVA/MMT K10 clay/AgNPs films and used them to improve the shelf-life of chicken sausages. The results showed that this type of nanocomposite packaging system is suitable for sausages, inhibiting the growth of total aerobic bacteria [10]. 

Starch/cellulose nanocrystal/grape pomade extract ‘Viognier’, nanocomposite films exhibited the highest antibacterial effect against *L. monocytogenes* inoculated on the ready-to-eat chicken meats during a 10-day storage period at 4 °C [11]. Strong antimicrobial properties were exhibited by chitosan–starch–cellulose nanofibrils films during storage of fresh beef sirloin [12]. The antimicrobial properties and high biocompatibility of chitosan/GO/TiO_2_ NPs nanocomposite films were observed when used in packaging strawberries and mangos. The results indicated that nanocomposite wrap effectively delays loss of weight in fruits, with weight loss below 5% after 7 days of storage, and prevents bacterial contamination in food products [13].

The influence of nanofillers on the properties of LDPE synthetic film was also studied. The use of CuNPs as an active additive to LDPE films positively influenced the shelf-life of Peda (Indian sweet dairy product) [151]. Luo et al. (2015) incorporated nano-SiO_2_ into LDPE films and used this type of packaging for storing chilled white shrimps. LDPE+nano-SiO_2_ are an active packaging material with antimicrobial and enzyme-inhibiting properties, in which shrimps remain fresh for 8 days [14]. Donglu and co-authors obtained a nanocomposite packaging material from LDPE and SiO_2_/TiO_2_ /Ag NPs, which regulated CO_2_ and O_2_ levels as well as inhibited the growth of microorganisms and scavenge ethylene during storage of mushrooms (*Flammulina velutipes*) [15].

Sodium CMC nanocomposite films with photoluminescent ZnS NPs exhibited a blue emission centered at 445 nm under UV light excitation, which can be easily incorporated into paper sheets in order to prepare protective paper, which can be used as a food packaging [255]. Rice straw nanofibrillated cellulose (NFC)/CHNPs nanocomposite films were used to coat baggasse paper. The coating of paper sheets confirmed improvement in paper sheet properties, improving the tensile strength, grease-proof properties, antibacterial activity, and reduced porosity [110]. El-Wakil and co-workers used gluten/CNC/TiO_2_ NPs nanocomposite films to coat paper sheets. The paper sheets coated with the aforementioned nanocomposite showed antimicrobial activity against *S. cervisiae*, *E. coli*, and *S. aureus* [256]. Nevertheless, further research is necessary to confirm the use of paper sheets coated with nanocomposite as an active packaging system, because there is still a lack of relevant in vivo studies.

In addition, the effectiveness of halloysite nanotubes as nanocontainers for active substances - sodium sorbate (a commonly used food preservative) - embedded in the zein matrix was tested. Preliminary studies have shown that it is possible to obtain an active packaging that has special packaging systems for the controlled release of active substances [257]. The presence of nanoclay in carnauba wax coatings enhanced the sensory acceptability and nutritional quality and effectively prevented the weight loss of ‘Valencia’ oranges (*Citrus sinensis* L. Osbeck) during storage at 4 °C for 8 weeks [258]. 

Currently, a large limitation in the commercial use of nanocomposite materials is the level of transition of nanomaterials from packaging to the food matrix. Although a number of studies have been carried out on the effects of using nanocomposites on prolonged storage of various types of food, consumers and legislation bodies expect confirmation of the safety of using nanocomposite packaging. One of the main concerns related to the use of nanocomposites is the emergence of new, diverse strains of bacteria and allergens, as well as the increased degree of adsorption of nanomaterials within the environment [67]. There are studies in which the migration of nanoparticles from food packaging has been studied [259,260]. Biodegradable starch/clay nanocomposite films could be used as food packaging owing to their low overall migration limit [260]. In addition, contact tests using food simulants are carried out to check the safety of the obtained nanocomposite materials. For example, the chitosan films with SeNPs met the required regulations and European directives on food packaging (EN1186-1, 2002), according to Commission Regulation No. 10/2011 [261].

### 4.2. Wound Dressing

The presence of ROS in wound often hampers wound healing. Cerium oxide nanoparticles (CeO_2_ NPs) with strong antioxidant properties may contribute to active ROS purification. One of the methods for the production of wound dressings is the electrospinning technique, which consists in obtaining fibers ranging from tens of nanometers to micrometers [262]. Naseri-Nosor and co-workers (2017) developed CeO_2_ NPs with poly (ɛ-caprolactone)/gelatin films as potential wound dressing material. After 2 weeks, wounds treated with PCL/gelatin+ 1.5 % CeO_2_ exhibited complete wound closure compared to control film without CeO_2_ [263]. 

To prevent damage caused by the removal of wound dressings, plasma treated with electrospun polycaprolactone was coated with gelatin–silver nanoparticle membranes using the multi-immersion technique. Compared to monolayer coating, this type of multilayer coating (six-time coated membrane) accelerated the healing of wounds by 3 days. Moreover, the membrane did not adhere to the wound during peeling off, preventing damage to the newly formed tissue [264]. The antibacterial chitosan/poly(vinyl pyrrolidone)(PVP)/nanocellulose films exhibited good compatible property and low cytotoxicity with blood. Moreover, this type of nanocomposite films could be used as wound dressing material due to their swelling, thermal, and mechanical properties [71]. The reinforcement of nanostarch in chitosan/PVP/1%-Stearic acid films caused the enhancement in the healing effect of albino rats [71].

Antimicrobial agent, chlorhexidine (CLX), was intercalated between the layers of MMT in the chitosan films. The chitosan films with MMT containing 5% CLX were non cytotoxic, whereas in the case of 1% CLX films only film with neat CLX resulted in cytotoxicity. The localized and prolonged release of CLX on the chitosan-MMT matrix could be a good solution in the area of wound dressings [265]. Cacciotti and coworkers (2018) designed poly(lactic) acid fibrous membranes with addition of H_2_S slow releasing donors extracted from garlic. The obtained antibacterial fibrous scaffolds/patches not only have the ability to stimulate cMSC (cardiac mesenchymal stem cells) proliferation, but also reduce oxidative damage [266].

Currently, nanotechnology is continuing to gain interest in the field of skin regeneration. On a laboratory scale, nanofillers placed in a polymer matrix accelerate wound healing, due to their antibacterial activity and improved angiogenesis. There are several important issues regarding the use of nanofillers in the wound healing process. Above all is the issue of dressing stability, so that nanofillers do not form a covalent bond with the polymer chain. In addition, there is still little data which would study the long-term safety of the use of nanofillers. Therefore, the mechanism of nanofillers should be thoroughly studied in order to be able to manage the cell behaviors. Understanding the mechanism of nanofillers will create a broader perspective on the safety of their use, which is why the long-term control over the performance of nanofillers is required [267].

### 4.3. Drug and Enzyme Delivery System

A drug delivery system is based on the delivery of a pharmaceutical drug to a specific site of a disease with minimized toxicological risk. Nanotechnology supports this by producing nano-scale materials, which can be successfully used as carriers [268]. A gelatin film with magnetic nanoparticles can be used as a control system for drug release, where the use of a magnetic field is important. Research has confirmed that without a magnetic field, this type of film acts as a blockage of drug release, while under the influence of a magnetic field, it opens the diffusion pathways [207]. Javanbakht and Namazi (2018) obtained flexible nanocomposite CMC hydrogel films with graphene quantum dots and anticancer drug, doxorubicin, release property. This type of nanocomposite caused pH-sensitivity and consecutively prolonged the release of doxorubicin and showed non-significant toxicity against blood cancer cells (K562) [269].

Nanocomposite films not only can be carriers of drugs, but also for example, can transport proteolytic enzymes during the wine production process. Bromelain produced from pineapple stem is a proteolytic enzyme, which reduces the haze potential of white wines. Chitosan films with different nanoclays (MMT, sepiolite, and bentonite) were produced as supports for the covalent immobilization of proteolytic enzymes. The addition of these clays affected the amount of immobilized protein, which was higher for all the nanocomposite films compared to the clay-free sample (with the exception of modified bentonite). On the other hand, the addition of nanoclays negatively affected the catalytic properties of the immobilized protease [270].

### 4.4. Tissue Engineering Application

Tissue engineering relies on the regeneration of damaged tissues or organ reconstruction. One of the methods used in tissue engineering is the use of materials that act as implantable scaffolds. Nanotechnology proposes to create nanocomposites that offer different structures and properties [271].

Hassan and co-workers (2018) obtained nanocomposite scaffolds based on chitosan and CMC AgNPs decorated on carboxylated CNWs. Those scaffolds exhibited both mechanical properties, due to carboxylated CNWs addition, and antimicrobial properties due to AgNPs. This type of nanocomposite exhibited sufficient protein adsorption and mineralization capacity which can overcome bone related infections like osteomyelitis [272]. Selenium nanoparticles (SeNPs), with their excellent antioxidant activity were used as nanofiller in chitosan films to produce electrical conductivity in cardiac patches. Due to its electrical and mechanical properties, this innovative material can be used in cardiac tissue engineering [273]. Nanocomposite films, obtained from PVA and zirconium phosphate doped with Ca, Mg, and Ti nanoparticles were used for scaffold-guided tissue engineering application. It was observed that PVA films with Ca- doped ZrP nanoparticles had increased biological activity, whereas PVA films with Ti-doped ZrP nanoparticles had relatively higher mechanical properties [274]. Hydroxyapatite (HA) is the main component of inorganic bones and is a very good material for the development of internal scaffolds for bone repair [275]. HA nanoparticles promote adhesion, proliferation, and osteogenic differentiation of osteoblast-like cells therefore they were used in preparation of chitosan-HA nanocomposite films for bone tissue engineering application [276]. Arumugam and co-workers (2019) developed novel, porous, mechanically stable, hydrophilic nanocomposite polyvinylidene fluoride/poly(methyl methacrylate)/hydroxyapatite/TiO_2_ film scaffolds. The in vitro study confirmed its excellent osteocompatibility, so this type of nanocomposite material could be used as potential material for bone repair applications [277]. The poly(L-lactic acid)/Ca-deficient-hydroxyapatite mats obtained by the electrospinning method can be used as a potential scaffold for bone marrow mesenchymal stem cell culture [278]. D’Angelo and coworkers (2012) investigated potential responses of multipotent (human-bone-marrow-derived mesenchymal stem cells) and pluripotent stem cells (murine-induced pluripotent stem cells and murine embryonic stem cells) to poly(L-lactic acid)/Ca-deficient-hydroxyapatite mats. They observed that the osteogenic differentiation effect of mats was not dependent on the type of stem cells [279].

In another work, HA was added to GO-chitosan films by using the layer-by-layer technique along with the biomimetic mineralization procedure. The results indicate the potential use of aspirin-loaded GO-chitosan-HA films in the field of bone tissue engineering [280].

### 4.5. Other Applications

Nanocomposite materials are also used in other industries. The addition of ZnO nanoparticles into chitosan/PVA films indicate their efficient removal of AB 1 azo dye (dye component) from aqueous solution [214]. The MWCNT was functionalized by valine and starch to improve the compatibility of MWCNT with chitosan/PVA matrix. The results indicate that this nanocomposite film is a very interesting adsorbent for the removal of Cd(II) from aqueous solutions [239]. 

Due to their inherent optical properties, the presence of carbon quantum dots in CMC films effectively caused ultraviolet light to convert into blue light. Transparent sunlight conversion CMC film with carbon quantum dots could be used in agriculture planting [281]. Mitta et al. (2018) obtained DNA thin films incorporated with AuNPs to demonstrate efficient UV photodetectors. Due to the presence of Au NPs, photodetectors were characterized by stability and durability [282].

## 5. The Future of Nanomaterials

According to the regulations of the European Commission, nanomaterials are covered by a regulatory framework. The presence of nanomaterials, including in food and cosmetics, must be indicated on the label in the list of ingredients. In addition, consumers must have access to online resources and databases on nanomaterials [253]. Due to the promising properties of nanomaterials, increasingly more new materials are appearing and the assessment of their risks requires an individual approach to each nanomaterial. There are major concerns that the high surface-to-volume ratio of nanomaterials may result in their having higher reactivity and potential toxicity. Due to the concerns about the safety of using nanomaterials, further research is necessary to give an unambiguous answer as to whether and which nanomaterials can be a viable alternative to traditional materials for application in many different areas [226,253].

## Figures and Tables

**Figure 1 polymers-11-00675-f001:**
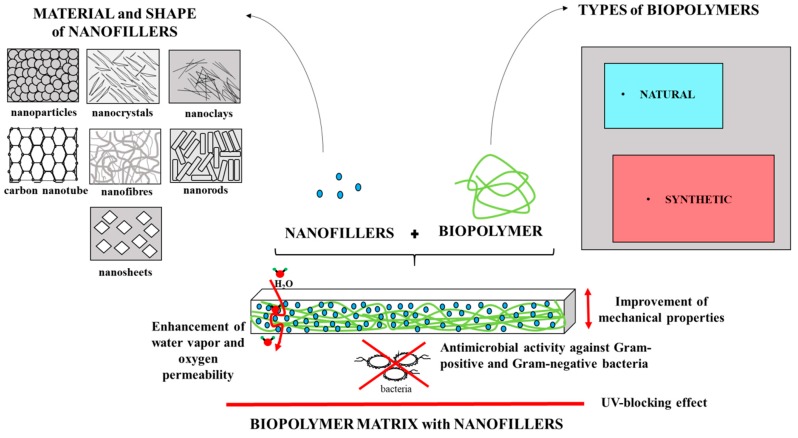
Schematic preparation of nanocomposite films and their functional properties.

**Figure 2 polymers-11-00675-f002:**
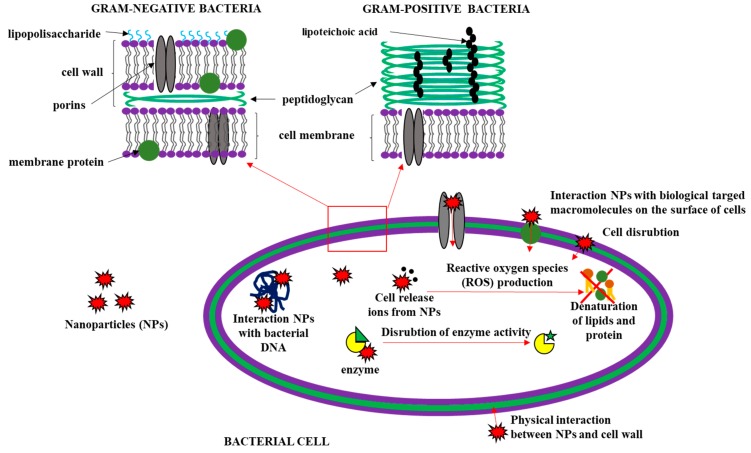
Potential mechanisms of antimicrobial activities of nanoparticles.

**Table 1 polymers-11-00675-t001:** Advantages and disadvantages of biopolymer films.

Type of Biopolymer Film	Advantages	Disadvantages
cellulose-based films	tasteless, odorless, resistant to oil and fat, hydrophilic nature [27]; thermal and chemical stability [39]	hardly dissolves or melts due to high crystallinity [40]; non antimicrobial activity [41]
chitin and chitosan-based films	good CO_2_ barrier properties, antimicrobial activity [42]	non antioxidant and antifungal activity [43]; limited oxygen and water impediment ability [44]
starch-based films	odorless, tasteless, good O_2_ and CO_2_ barrier properties [45]	poor water vapor barrier [46] and tensile properties [47]
pectin-based films	excellent oxygen barring capacity [48]	high water vapor permeability [49]; poor mechanical performance [48]
pullulan-based films	heat-sealable [50]; highly impermeable to both oil and oxygen [51]; excellent mechanical properties and a low permeability to oil and oxygen [52]	low solubility [50]; hydrophilic nature [52]
alginate-based films	good water solubility, gel ability, and film-forming properties [53]	insufficient mechanical properties and poor water resistance [54]
gelatin-based films	good mechanical and barrier properties [55]	low water vapor permeability [56]
whey protein-based films	excellent barrier properties to aroma compounds and oils [27]	hydrophilic nature so it has limitation to moisture [27]
lipids-based films	excellent barriers against moisture migration [27]	damage the appearance and gloss of the coated food products [57]
bacterial cellulose-based films	flexibility and excellent mechanical properties [58]	insoluble in water [59]
PCL- based films	high mechanical strength, biocompatibility, processability, and permeability [60]	highly hydrophobic and crystalline [61]
PLA-based films	environmental friendliness, good transparency, and biological compatibility [62]	high hardness, and brittleness, low strength, and poor thermal stability [63]
PGA-based films	high mechanical strength [64]	high degree of crystallinity, a high melting point, and it is insoluble in common organic solvents [65]
PU-based films	favorable processability, versatile structure–property relationships, and excellent elasticity [66]	low water resistance and hardness [66]

**Table 2 polymers-11-00675-t002:** Recent examples of nanofillers properties.

Type of Nanofillers	Properties Added to the Film	Reference
**Clay Nanofillers**		
MMT, Hal etc.	UV shielding properties	[69]
	Good mechanical stability	
	Thermal stability	[70]
**Organic Nanofillers**		
Nanocellulose	Blood compatibility	
	Antibacterial effect	[71]
	Thermal stability	[72]
	Good mechanical stability	
	Low cytotoxicity	[73]
Chitosan nanoparticles	Biocompatibility	
	Biodegradability	[74]
	Low toxicity	[75]
	Antimicrobial activity	[76]
**Inorganic Nanofillers**		
AgNPs	Antimicrobial effect	[77]
	UV shielding properties	[78]
	Antioxidant activity	[79]
	Photocatalytic effect	[80]
SeNPs	Antimicrobial effect	[81]
	Antioxidant activity	[82]
CuNPs	Antimicrobial effect	[83]
	UV shielding properties	[84]
SNPs	Antimicrobial effect	[85]
TiO_2_ NPs	Antifouling effect	[86]
	Antimicrobial activity	[87]
	Photocatalytic activity	[88]
	UV shielding properties	[69]
ZnO NPs	Antifungal effect	
	UV shielding properties	[89]
	Antimicrobial effect	[90]
	Dielectric properties	
	Electromagnetic shielding	
	Thermal conductivity	[91]
CeO_2_	Antimicrobial effect	
	UV shielding properties	
	Flame retardancy	
	Wrinkle resistance	[92]
**Carbon Nanofillers**		
Graphene, graphene oxide, etc.	Lightweight	
	Processing benefits, flexibility, resistance to corrosion	
	Extraordinary electrical, mechanical, and thermal properties	[93]

**Table 3 polymers-11-00675-t003:** Recent examples of nanocomposite films.

Nanofiller	Polymer	The Effect of Nanofiller Addition	Reference
**Metal Nanostructures**			
AgNPs	gelatin	• Improvement of antibacterial effect from 0 up to 14 mm of inhibition zone against *S. typhimurium*, *B. cereus, L. monocytogenes, E. coli*, and *S.aureus* • Reduction of TS and YM by up to ~25 % and ~36%, respectively, depending on AgNPs concentration • No changes in EAB, WVP, MC, and WCA	[199]
AgNPs	chitosan-gelatin	• Reduction of TS (~27%) and improvement of EAB (~34%) • Increased the shelf-life of red grapes on which the film was applied	[200]
AgNPs	chitosan	• Improvement of antimicrobial activity against *P. aeruginosa* (to ~28 mm of inhibition zone), *S. aureus* (to ~37 mm), and MRSA (to 24.73 mm), depending on AgNPs concentration	[201]
AgNPs	chitosan/cellulose	• Improvement of antimicrobial activities against *S. aureus* (~0.8 mm of inhibition zone) and *E. coli* (~1.2 mm)	[202]
AgNPs	chitosan/PVA	• Improvement of antioxidant activity by up to ~33% (DPPH radical scavenging activity), up to ~37% (ferric reducing ability) and up to ~31% (β-Carotene bleaching), depending on AgNPs concentration • Low toxicity • Improvement of antimicrobial activity against *S. areus (* increase by ~1 mm of inhibition zone)*, B. cereus* (~8 mm)*, M. luteus* (~2 mm)*, S.enterica* (~3 mm)*, E. coli* (~3 mm), and *P. aeruginosa* (~2 mm)	[203]
AgNPs AuNPs	chitosan	• Improvement of antimicrobial activity • AuNPs has better activity against *A. niger* than AgNPs (from 0 to 25 mm of inhibition zone) • AgNPs has better activity against *C. albicans* than AuNPs ( from 6 to 19 mm of inhibition zone) • No significant difference between antimicrobial activity of AuNPs and AgNPs against *S. aureus* and *P. aeruginosa*	[204]
AgNPs SeNPs	furcellaran	• Enhancement of MC (with AgNPs ~11.5% and with SeNPs ~14%), WS, EM (with AgNPs and with SeNPs ~10%), but reduction in SR (with AgNPs ~13% and with SeNPs ~20%) • AgNPs improved the UV-blocking effect • No changes in EAB • SeNPs improved antimicrobial activity against *E. coli* (SeNPs from 0 up to ~38 mm of inhibition zone; AgNPs from 0 to ~10 mm)*, S. aureus* (SeNPs from 0 up to ~22 mm), and MRSA (SeNPs from 0 up to ~26 mm)	[81]
SNPs	chitosan	• Increment of TS (by up to ~18%), EM (by up to ~18%) and WCA (by up to ~6%) • Reduction of EAB (by up to ~39%), WVP (by up to 14%) • Enhancement of thermal stability • Antimicrobial activity against *L. monocytogenes* (complete destroy after 12 h) and *E. coli* (complete destroy after 6 h)	[85]
Lignin capped AgNPs	agar	• Enhancement of UV screening effect • Improvement of TS (by up to ~23%) and YM (by up to ~13 %) • No changes in EAB • Reduction of WVP (by up to ~22%), WCA (by up to ~9%), WS (by up to ~16%), and SR (by up to ~50%) • Improvement of antimicrobial activity against *E. coli* (complete destroy after 6 h) and *L. monocytogenes* (complete destroy after 12 h)	[150]
Ag-Cu NPs	gelatin	• Strong UV screening effect • Improvement in TS (by up to ~49%) but reduction in EAB (by up to ~40%) • Strong antimicrobial activity against *S. thyphimurium* (from 3.5 to 7 log) and *L. monocytogenes* (from 0.5 to 3 log) • Enhancement of thermal stability	[173]
AgNPs inside gelatin-montmorillonite (AgM)	cellulose acetate	• Improvement of UV barrier, TS (by up to ~6%) and EM (by up to ~18%), but reduction in EAB (by up to~50%) • Reduction of OP (by up to ~14%) • Enhancement of thermal stability • Improvement of antimicrobial activity against *E. coli* (from 0 to 36 mm of inhibition zone)*, S. aureus* (from 0 to 34 mm)*, Salmonella* (from 0 to 32 mm)*, Psuedomonas* (from 0 to 35 mm)*, A. niger*, and *A. flavus*, depending on the AgNPs and thymol concentration	[205]
AgNPs chitin nanofiber	chitosan	• AgNPs reduced TS, YM, and color properties • CHNF improved WS, SR, WVP, TS, and YM	[144]
AgNPs nanocellulose	PVA	• AgNPs enhanced TS and EAB and reduced WVP • Enhancement of thermal stability • Improvement of antimicrobial activity against *E.coli* and MRSA	[206]
**Metal Oxide Nanostructures**			
Magnetite nanoparticles	gelatin	• Enhancement of thermal stability	[207]
Magnetic nanoparticles	chitosan	• Improvement of TS (by up to ~37%) and EAB (by up to ~18%)	[208]
TiO_2_ NPs	gelatin	• Enhancement of TS (by up to ~60%) and EAB (by up to ~48%) • Improvement of barrier properties against UVC light • Irradiation of the film with UV-A light (365 nm) resulted in the most effective antibacterial activity against *E. coli*	[209]
TiO_2_ NPs	gelatin–agar	• Improvement of TS (by up to ~29%) and reduction in EAB (by up to ~22%) • Increment of WS (by up to ~4%) and MC (by up to ~10%) • Reduction of WVP (by up to ~32%) • UV blocking effect	[146]
TiO_2_ NPs	chitosan	• Improvement of antimicrobial activity against bacteria (*S. aureus, E. coli, S. typhimurium*, and *P. aeruginosa*) and fungi (*Aspergillus spp.* and *Penicillium spp.*) • Enhancement of TS (by up to ~56%), WVP (by up to ~22%) and ethylene photocatalytic degradation properties • Reduction of EAB (by up to ~10%)	[210]
TiO_2_ NPs	potato starch	• Reduction of WS (by up to ~9%), MC (by up to ~11%) and WVP (by up to ~35%) • Improvement of TS (by up to ~45%) and reduction of EAB (by up to ~28%) • Improvement of UV-blocking effect	[121]
CuO NPs	PVA–gelatin	• Improvement of UV screening effect	[123]
ZnO NPs	gelatin	• Improvement of UV screening effect and thermal stability • Reduction of EM (by up to ~82%) and TS (by up to ~72%) • Increment of WVP (by up to ~99%), MC (by up to ~29%), WCA (by up to ~20%) • Improvement of antimicrobial activity against *E. coli* (from 9 to 5 log) and *L. monocytogenes* (from 9 to 1 log)	[160]
ZnO NPs	chitosan/CMC	• Improvement of shelf life of white soft cheese on which the film was applied • Improvement of antibacterial activity against bacteria *S. aureus* (from 5 to 11 mm of inhibition zone) *P. aeruginosa* (from 3 to 11 mm), *E. coli* (from 3 to 9 mm), and fungi *C. albicans* (from 3 to 15 mm) • Improvement of TS (by up to ~85%)	[211]
ZnO NPs	chitosan	• Improvement of TS but reduction of EAB • Improvement of antimicrobial activity against *E. coli* (3.4 log reduction after 0.5 h) and *S. aureus* (4 log reduction after 0.5 h) • Biocompatibility and nontoxicity	[212]
ZnO NPs	mahua oil-based polyurethane/chitosan	• Improvement of TS (by up to ~56%) but reduction of EAB (by up to ~20%) • Reduction of OP (by up to ~3%) and WVP (by up to ~37%) • Improvement of antimicrobial activity against *E. coli* (~25 mm) and *S. aureus* (~20 mm) • UV-screening ability and biodegradation • Non-cytotoxic	[213]
ZnO NPs	PLA	• Improvement of TS (by up to ~37%), WVP (by up to ~31%) and UV-light barrier properties • Reduction of EAB (by up to ~10%) • Improvement of antibacterial activity against *E. coli* (from 10 to 3.5 log after 12 h) and *L. monocytogenes* (from 12 to 8 log after 12 h)	[31]
ZnO NPs	chitosan/PVA	• Improvement in photoluminescent properties and thermal stability	[214]
SnO_2_ NPs	CMC	• The choice of nanocomposite preparation procedure caused four different morphologies of SnO_2_ NPs (microcube, nanosphere, olive-like and nano-flower) which had different effects on thermal stability of CMC	[125]
ZnO NPs CuO NPs	carrageenan	• ZnONPs strongly improved antimicrobial activity against *E. coli* and *L. monocytogenes*, UV-blocking effect and thermal stability • Reduction of TS (by up to ~55%) and EM (by up to ~26%) depending on the CuO NPs and ZnO NPs concentration and the ratio of nanoparticles • Improvement of EAB	[215]
Fe_2_O_3_ NPs	cellulose	• Improvement of TS (by up to ~10%) and YM (by up to ~15%) and thermal stability	[216]
MgO NPs	PLA/polyethylene glycol	• Improvement of EAB (by up to ~86%) but reduction of TS (by up to ~64%) • Enhancement of optical properties	[217]
α- Fe_2_O_3_ NPs FeNPs	chitosan/PVA	• Improvement of magnetic properties • FeNPs decreased TS and EM depending on the composition of chitosan and PVA • Fe_2_O_3_ NPs increased TS and EM depending on the composition of chitosan and PVA • FeNPs and Fe_2_O_3_ NPs caused reduction in EAB	[33]
ZnO nanorod nano-kaolin	semolina	• Reduction of OP (by up to ~34%), MC (by up to ~64%) and WS (by up to ~56%), depending on the ratio ZnO nanorods/nano-koalin • Improvement of UV barrier properties and antimicrobial activity against *E. coli* (from 0 to ~3 mm)	[218]
ZnO nanorods	starch/gelatin	• Reduction of OP (by up to ~61%) • Improvement of TS (by up to ~30%) but decrement of EAB (by up to ~44%) • Enhancement of UV barrier properties	[174]
ZnO nanorods	gelatin/clove essential oil	• Reduction of TS (by up to ~61%) and increment of EAB (by up to ~155%) and OP (by up to 98%) • Increment of oxygen and UV barrier property • Improvement of antimicrobial activity against *L. monocytogenes* (from 10 to 0 log after 7 days) and *S. typhimurium* (from 10 to 0 log after 7 days) • Improvement of shelf-life of peeled shrimps	[219]
ZnO nanorods	gelatin	• Decrement of hydrophobicity and moisture contents • Reduction of WVP (by up to ~80%) • Improvement of UV-blocking effect and antimicrobial activity against *S. aureus* (from 0 to 80 mm^2^ of inhibition zone)	[185]
ZnO nanorods	soybean polysaccharide	• Reduction of WVP (by up to ~36%) and OP (by up to ~43%) • Decrement of TS (by up to ~18%) and increment of EAB (by up to ~41%) and heat seal strength (by up to ~29%) • Improvement of antimicrobial activity against *E. coli* (from 7 to 5 log after 12 h) and *S. aureus* (from 6 to 1 log after 12 h) and UV-blocking effect	[220]
ZnO nanorods	PVA/CMC	• Improvement of dielectric properties	[221]
**Cellulose Nanostructures**			
cellulose nanocrystals	carrageenan	• Improvement of TS (by up to ~70%) and toughness (by up to ~10%) parameters • Reduction of EAB (by up to ~25%) • Enhancement of thermal stability	[179]
rice cellulose nanocrystals	chitosan/PVA	• Improvement of TS (by up to ~75%) and EM (by up to ~98%) • Reduction of EAB (by up to ~43%) • Enhancement of thermal stability • No changes in antifungal against *C. gloeosporioides* and *L. theobromae* and antimicrobial against *S. mutans, S. aureus, E. coli*, and *P. aeruginosa* activities	[222]
cellulose nanocrystals	chitosan	• Improvement of mechanical properties (by up to ~44%) and thermal stability	[106]
flax cellulose nanocrystals	chitosan	• Improvement of TS (by up to ~24%), EAB (by up to ~22%) and YM (by up to ~140%) • Reduction of WVTR (by up to ~11%) and increment of WVP (by up to ~85%) • Enhancement of antimicrobial activity against *P. aeruginosa, E. faecalis, L. monocytogenes, E. coli,* and *S. aureus* (from 6.31 to 16.05 mm of inhibition zone)	[223]
bacterial cellulose nanocrystal	PVA	• Improvement of TS, YM, and toughness depending on the presence of glycerol, boric acid, and BCNC • No changes in EAB	[105]
cellulose nanowhiskers	chitosan	• Increment of YM but reduction of TS and EAB • Enhancement of thermal stability	[224]
nanocrystalline cellulose	chitosan/guar gum	• Improvement of the shear viscosity of the suspensions • Reduction of air permeability (by up to ~53%) and EAB (by up to ~53%) • Improvement of YM (by up to ~86%) and TS (by up to ~75%)	[225]
sugar palm nanocrystalline cellulose	sugar palm fibre	• Reduction of MC (by up to ~19%), WS (by up to ~56%) • Improvement of WVP (by up to ~18%) and thermal stability	[153]
bacterial cellulose nanocrystals and AgNPs	chitosan	• Improvement of UV barrier properties • Enhancement of physical, mechanical properties and in thermal stability, depending on the concentration and ratio of AgNPs and BCNC • Improvement of antimicrobial and antifungal activity (from 0 to 96 mm^2^ of inhibition zone depending on the concentration and ratio of AgNPs and BCNC)	[184]
cellulose nanocrystals	chitosan alginate κ-carrageenan	• Improvement of YM, TS, and toughness in every type of film • Reduction in parameter of EAB	[178]
cotton linter cellulose nanofibril	CMC	• Improvement of TS (by up to ~23%) and EM (by up to ~28%) but reduction of EAB (by up to ~26%) • Enhancement of thermal stability • Reduction of WCA (by up to ~39%) • No changes in WVP	[226]
celullose nanocrystals	CMC	• Enhancement of thermal stability • Improvement of TS (by up to ~74%) and EM (by up to ~129%) and reduction of EAB (by up to ~47%)	[227]
cellulose nanofibers	soy protein	• Improvement of TS (by up to ~400%) and YM (by up to ~767%) • Reduction of EAB (by up to ~56%) • No effects of WVP	[228]
cellulose nanocrystals	cassava starch	• Reduction of WVP (by up to ~43%), oil permeability (by up to ~42%) and MC • Improvement of TS (5.6 times higher than cassava starch films) • Increment of WS	[229]
cellulose nanocrystals	PVA/CMC	• Enhancement of TS (by up to ~83%) and EM (by up to ~147%) • Reduction of WVP (by up to ~82%)	[2]
crystalline nanocellulose	CMC/chitosan	• Enhancement of barrier against grease and oil • Improvement of TS (52% higher than CMC/chitosan films), YM, and WVP (by up to ~38%) • Reduction of strain at break	[230]
Cellulose nanofibers TiO_2_ NPs	whey protein	• TiO_2_ increased the water resistance • TiO_2_ and CNFs improve mechanical properties and the values depended on the concentration of nanofillers • TiO_2_ enhances antimicrobial activity against *L. monocytogenes* and *S. aureus* and antioxidant properties	[87]
licorice residue nanocellulose	soy protein isolate	• Improvement of TS but reduction of EAB • Reduction in WVP (by up to ~27%) and OP (by up to ~55%) • UV-blocking effect	[152]
**Nanoclays**			
betonine nanoclays	chitosan/PVA	• Reduction of WVP (by up to ~69%) and TS (by up to ~30%) • Improvement of thermal stability • Enhancement of antibacterial properties against *E. coli* (efficiency 48.50 %)*, P. aeruginosa* (efficiency 40%)*, S. aureus* (efficiency 8%)	[231]
cloisite Na+ nanoclays	agar	• Improvement of TS (by up to ~31%) • Reduction in WVP (by up to ~50%), WCA (by up to ~10%), and WS (by up to ~23%)	[155]
halloysite nanotubes	chitosan/starch	• Reduction of WS • Improvement of water absorption capacity, porosity, folding strength, and WVTR • Improvement of impermeability to bacteria (the microbial penetration test)	[145]
hallosite nanotubes with metal ions (Ag, Zn, Cu)	CMC	• Improvement of WVP and thermal stability • Enhancement of antimicrobial activity against *L. monocytogenes* and *E. coli* • Improvement in TS and EM but reduction of EAB (all results were depending on the type of metal ions in hallosite nanutubes)	[232]
nanoclays: Na+ montmorillonite halloysite Nanomer® I.44 P	fenugreek seed gum	• Enhancement of thermal stability • No influence on antimicrobial activity against *E. coli*, *S. aureus*, *B. cereus* • Antimicrobial activity against *L. monocytogenes* (all results were depending on the type of nanoclays)	[168]
halloysite nanotubes loaded with the essential oil	pectin	• Reduction in thermal stability and EAB • Improvement in TS and EM and surface hydrophobicity	[233]
halloysite nanotubes ZnO NPs	alginate	• Improvement of TS (by up to ~12%), WVP (by up to ~27%), and WCA (by up to ~28%) • No changes in EAB (by up to ~6%) and EM (by up to ~7%) • Improvement in UV-blocking effect and antimicrobial activity against *E. coli* (from 7 to 0 log after 3 h) and *L. monocytogenes* (from 6 to 0 after 9 h)	[95]
Cloisite 30B AgNPs	gelatin	• Improvement of optical properties • AgNPs improved TS while nanoclay increased EAB • Enhancement of antimicrobial activity against *E. coli* and *L. monocytogenes*	[234]
bismuth tungstate/ TiO_2_ NPs (Bi_2_WO_6_-TiO_2_)	starch	• Improvement of TS (by up to ~233%) and photocatalytic activity • Reduction of EAB (by up to ~15%)	[235]
halloysite nisin	starch	• HTN improved TS and YM • Nisin reduced TS and YM but increased antimicrobial properties against *L. monocytogenes, C. perfringens*, and *S. aureus* • No changes in WS	[236]
calcium montmorillonite	carboxymethyl starch	• Improvement of TS (by up to ~500%), YM (by up to ~1733%), and WCA (by up to ~53%) • Reduction of EAB (by up to ~49%), WS (by up to ~4%)	[237]
montmorillonite cellulose NPs	alginate	• Reduction of WVP • CNC increased the TS and EAB, while MMT in high concentration reduced the TS	[238]
montmorillonite − CuO nanocomposites	chitosan	• Enhancement of TS, EAB, WVP, and OP • Reduction of WS and UV transition • Improvement of antimicrobial activity against *E. coli, P. aeruginosa, S. aureus, B. cereus* (all results were depending on the concentration and ratio of MMT and CuONPs	[169]
montmorillonite ZnO nanopowders	cationic starch	• MMT reduced of WVP and UV light transmittance • ZnO improved of WVP and UV light transmittance • Improvement of TS and optical properties, but reduction of EAB	[166]
sodium montmorillonite nanoclay ZnO	CMC	• Reduction in WVP • Addition of ZnO NPs increase UV-blocking effect • Addition of ZnO NPs enhance antimicrobial activity against *E. coli* and *S. aureus*	[154]
**Carbon Nanostructures**			
multi-walled carbon nanotube-Valine	chitosan/PVA	• Improvement of thermal stability	[239]
carbon nanotubes AgNPs	chitosan	• Enhancement of TS (by up to ~131%), EAB (by up to ~18%) and toughness (by up to 125%) • Reduction of dielectric properties	[240]
graphene oxide	cellulose carbamate	• Improvement of thermal stability • Improvement of TS (by up to ~100%) and reduction of EAB (by up to ~64%)	[129]
graphene nanoplatelets	CMC	• Reduction of ultimate tensile strength (by up to ~50%) but increment of strain to break (by up to ~66%) • Enhancement of UV-blocking effect • Improvement of water repelling nature	[241]
reduced graphene oxide	sodium CMC	• Improvement of TS (by up to ~73%) and YM (by up to ~132%)	[177]
graphene oxide	amylose	• Enhancement of stability in acidic and alkaline solutions • Improvement of TS (16.5 times higher than amylose films) • Reduction of the MC and UV transmittance	[242]
graphene oxide reduced graphene oxide	sodium CMC/silk fibroin	• Improvement of thermal stability	[131]
**Other Nanostructures**			
melanin nanoparticles	carrageenan	• Increment of TS (by up to ~27%), EAB (by up to ~25%), WVP (by up to ~25%) and WCA (by up to ~25%) • Reduction of YM (by up to ~38%) • Enhancement of antioxidant activity (by up to ~962%-DPPH method and by up to ~559%-ABTS method) and thermal stability • Improvement of antimicrobial activity against *L. monocytogenes* and *E. coli*	[243]
ZnS NPs	chitosan/PVA	• Reduction of WS and SR • Improvement of TS and thermal stability and reduction of EAB (all results were correlating with the type of plasticizer)	[244]
guar gum benzoate NPs	gelatin	• Improvement of antimicrobial activity against *E. coli* and *S. aureus* • Enhancement of thermal stability • Improvement of TS (by up to 67%) and YM (by up to ~550%) • Reduction in EAB (by up to ~63%)	[245]
chitosan NPs	rice straw nanofibrillated cellulose	• Improvement of TS (by up to ~40%) and YM (by up to ~42%) but reduction of EAB (by up to ~94%) • Enhancement of antimicrobial activity against bacteria (*S. aureus, E. coli),* and yeast (*S. cervisiae*) • Reduction in porosity • No changes in WVP	[110]
chitosan NPs	tara gum	• Improvement of TS and reduction of EAB • Reduction of WS (by up to ~74%), WVP (by up to ~23%) and MC (by up to ~24%) • Antimicrobial activity against *E. coli* (from 0 to 87.32 mm^2^ of inhibiton zone) and *S. aureus* (from 0 to 111.71 mm^2^)	[246]
chitosan NPs	PVA/mulberry extract	• Improvement of TS but reduction of EAB	[111]
lignin NPs	chitosan PVA chitosan/PVA	• Improvement of TS and YM of PVA films • Enhancement of thermal stability of chitosan, PVA, and chitosan/PVA films • Improvement of UV barrier properties of tested films • Enhancement of antioxidant activity of chitosan films with LNP • Improvement of antimicrobial activity against *Erwinia carotovora* subsp. *carotovora* and *Xanthomonas arboricola* pv. *Pruni*	[247]
chitosan/gallic acid NPs	konjac glucomannan	• Improvement of UV barrier properties • Enhancement of TS (by up to ~43%) but reduction of EAB (by up to ~16%) and WVP (by up to ~33%) • Improvement of antimicrobial activity against *S. aureus* (from 0 to 20 mm of inhibition zone) and *E. coli* (from 0 to 12 mm)	[167]
chitin nanofiber	gelatin/CMC	• Reduction of WS, SR and WVP • Improvement of TS	[109]
chitin nanowhiskers	maize starch	• Improvement of TS (by up to ~125%) and thermal stability but reduction of EAB (by up to ~37%) and WVP (by up to ~58%) • Enhancement of antimicrobial activity against *E. coli* and *L. monocytogenes*	[143]
oxidized chitin nanocrystals	CMC	• Improvement of TS (by up to ~88%) and EM (by up to ~244%) • Reduction of WVP (by up to ~10%), WCA (by up to ~14%) and EAB (by up to ~65%)	[248]
chitin nanowhiskers /hybrid ZnO-Ag NPs	CMC	• Enhancement of thermal stability and UV-barrier property • Improvement of TS (by up to ~32%) and YM (by up to ~101%) but reduction of EAB (by up to ~34%) • Enhancement of antimicrobial activity against *E.coli* (from 6 to 0 log after 6h) and *L, monocytogenes* (from 7 to 4 log after 9 h) • Reduction of WVP (by up to ~23%)	[249]
pullulan	lysozyme nanofibers	• Improvement of YM (by up to ~48%), TS (by up to ~7%) but reduction of EAB (by up to ~80%) • Enhancement of antioxidant activity (from 0 to ~80% DPPH method) and antimicrobial activity against *S. aureus*	[250]
chitosan	nanocrystalline erbium doped hydroxyapatite	• Improvement in antimicrobial activity against *E. coli* and *S. aureus*	[251]
potato starch tapioca starch chitosan	turmeric nanofiber	• Improvement of TS, YM, and thermal stability • Reduction of EAB • Antimicrobial activity against *B. cereus, E. coli, S. aureus*, and *S. typhimurium* (the values were depending in the type of biopolymer)	[252]
maltodextrin	polyvinyl acetate NPs	• Improvement of TS (by up to ~106%)	[165]

Abbreviations: NPs—nanoparticles; CMC—carboxymethyl cellulose; PVA—poly(vinyl) alcohol; PLA—poly(lactic acid); TS—tensile strength; EAB—elongation at break; YM—Young’s modulus; EM—elastic modulus; WVP—water vapor permeability; OP—oxygen permeability; WS—water solubility; SR—swelling ratio; MC—moisture content; WCA—water contact angle.

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
