# Peer review of "The Effect of Nanofillers on the Functional Properties of Biopolymer-Based Films: A Review"

_polymers, 2019, doi:10.3390/polym11040675_

Round 1
Reviewer 1 Report
General comments
The submitted paper consists in a review about biocomposites, based on biopolymeric matrix and nanofillers, in order to evidence the influence of the used nanofillers on the functional properties of the final composite.
The topic of the present work is worthy of investigation, and well fits the aim and scope of Polymers. However, the Authors should better organize the present review, evidencing that the main aim is to provide an overview about biocomposites for food packaging applications and that examples of other potential applications are only partially reported in addition. It is not so clear and evident.
Table 1 should be directly included within the manuscript and not as supplementary material. It is strongly suggested to add more tables with collected data e literature references, in order to provide a complete vision to the readers.
Moreover, a deep English grammar and language revision is strongly recommended.
More details and specific remarks and suggestions are reported below point by point.
Abstract
The Abstract has to be properly expanded, adding some examples of biopolymers and nanofillers, better contextualizing the manuscript topic.
Keywords
The chosen keywords (i.e. biopolymer films; nanofillers; functional properties) sound appropriate, but they do not completely cover the review content. It is strongly suggested to add further ones, more specific.
1. Introduction
- The Introduction section should be expanded and improved, in order to evidence the aim of the present review. The Authors shoukd better evidence that their review is focused on biocomposites for food packaging applications, as evident form the incipit, and that they reported other potential applications, at the end of the review.
- The following period “Currently, a novel method to improve the properties of biopolymer films is the use of nanofillers, which can fulfill not only the reinforcing function but also could act as an active ingredient. In recent years, the concept of active agents for biopolymer films has received much more attention. Such active ingredients in biopolymer films can extend the shelf-life of food products, through exhibiting antimicrobial and/or antioxidant activities. The development of nanotechnology has led to the design of nanocomposite film materials in which nanofillers play an active role.” Has to be properly supported with suitable recent literature references, including “Eco-sustainable systems based on poly(lactic acid), diatomite and coffee grounds extract for food packaging, Intern J Biolog Macromolecules 112 (2018): 567-575.” and “Poly(lactic) acid fibers loaded with mesoporous silica for potential applications in the active food packaging, AIP Conference Proceedings 1738 (2016): 270018 (1-4).”
2. Types of Biopolymers and Nanofillers
- The following statement “This solution applies to the production of cheap, biodegradable packaging material, which is based on biopolymers and nanofillers.” is not totally true, since another drawback of these materials consists in the high cost with respect to the traditional polymers.
- The sentence “Blending the different biopolymers together is one of the promising methods of improving the functional properties of biopolymer-based films.” Has to be supported with suitable literature references, including “Electrospun PHBV/PEO co-solution blends: microstructure, thermal and mechanical properties, Materials Science and Engineering: C 33[3] (2013): 1067–1077” and “Tailoring the properties of electrospun PHBV mats: co-solution blending and selective removal of PEO, European Polymer Journal 49 (2013) 3210–3222”.
1.1. Types of biopolymers matrix
- Firstly, please replace 1.1 with 2.1.
- Moreover among the synthetic biopolymers, polycaprolcatone has to be included, since it is widely employed. Thus proper references have to be reported, including “Electrospun poly(e-caprolactone)-based composites using synthesized β-tricalcium phosphate, Polymers for advanced technology 22[12] (2011): 1832–1841”.
2.2. Types of nanofillers
- It is not correct to classify nanoclays as organic fillers. Please correct it.
2.2.1. Organic nanofillers
- The following sentence “The different types of nanocellulose have been incorporated to many biopolymer films [29-31]” has to be corroborated with more recent literature references, including “Effect of silver nanoparticles and cellulose nanocrystals on electrospun poly(lactic) acid mats: morphology, thermal properties and mechanical behaviour, Carbohydrate Polymers 103 (2014): 22– 31.”.
- Similarly, the following consideration “Chitosan and chitin nanoparticles obtained respectively from chitosan or chitin have gained attention as nanofillers, due to their attractive surface area, biocompatibility, non-toxicity and film forming ability [32-35].“ has to be supported with more recent literature references, including ”Biodegradable zein film composites reinforced with chitosan nanoparticles and cinnamon essential oil: physical, mechanical, structural and antimicrobial attributes, Colloids and Surfaces B: Biointerfaces 177(2019): 25-32.
3. The Effects of Nanofillers on the Functional Properties of Biopolymer-Based Films
A paragraph about the effect on the gas barrier properties has to be added, being one of the most important requirements in the case of films for food packaging applications.
3.2. The effects of nanofillers on the mechanical properties of polymer-based films
- Concerning the effect of CNC and AgNPs, the following paper “Effect of silver nanoparticles and cellulose nanocrystals on electrospun poly(lactic) acid mats: morphology, thermal properties and mechanical behaviour, Carbohydrate Polymers 103 (2014): 22– 31.”. should be described and discussed, evidencing the synergic influence of different nanofillers if used simultaneously.
4. The Functional Application of Biopolymer-Based Films with Nanofillers
4.1. Wound dressing
For the wound dressing applications, it is suggested to discuss the results of the very recent paper “Hydrogen Sulfide-Releasing Fibrous Membranes: Potential Patches for Stimulating Human Stem Cells Proliferation and Viability under Oxidative Stress, International Journal of Molecular Sciences 19(8) (2018): 2368”.
4.4. Tissue engineering application
The following sentence “Hydroxyapatite (HA) nanoparticles promote adhesion, proliferation, and osteogenic differentiation of osteoblast like cells” needs to be supported with suitable literature references, including “Cationic and Anionic substitutions in hydroxyapatite, In: Handbook of Bioceramics and Biocomposites, Springer International Publishing 2016: 145-211.”, “Poly(L-lactic acid)/calcium-deficient nanohydroxyapatite electrospun mats for murine bone marrow stem cell cultures, Journal of Bioactive and Compatible Polymers26[3] (2011): 225-241” and “Tuning multi-/pluri-potent stem cell fate by electrospun poly(L-lactic acid)-calcium-deficient hydroxyapatite nanocomposite mats, Biomacromolecules 13 [5] (2012): 1350-1360”.
Author Response
General comments
The submitted paper consists in a review about biocomposites, based on biopolymeric matrix and nanofillers, in order to evidence the influence of the used nanofillers on the functional properties of the final composite.
Comments: The topic of the present work is worthy of investigation and well fits the aim and scope of Polymers. However, the Authors should better organize the present review, evidencing that the main aim is to provide an overview about biocomposites for food packaging applications and that examples of other potential applications are only partially reported in addition. It is not so clear and evident.
Response: The Abstract has been corrected and some parts have been rewritten. We have also modified the structure of the article so that the main emphasis is placed on the applications within the food industry and food packaging systems.
Comments: Table 1 should be directly included within the manuscript and not as supplementary material. It is strongly suggested to add more tables with collected data e literature references, in order to provide a complete vision to the readers.
Response: We have added Table 1 to the main text (its current number is Table 3), and have added more detailed information about the influence of the presence of nanofillers into biopolymer films. Additionally, we have added Table 1 containing information about advantages and disadvantages of some polymer films and Table 2 about the properties of example of nanofillers.
Comments: Moreover, a deep English grammar and language revision is strongly recommended.
Response: The manuscript has been checked by native speaker.
More details and specific remarks and suggestions are reported below point by point.
Abstract
Comments: The Abstract has to be properly expanded, adding some examples of biopolymers and nanofillers, better contextualizing the manuscript topic.
Response: The Abstract have been modified.
Keywords
Comments: The chosen keywords (i.e. biopolymer films; nanofillers; functional properties) sound appropriate, but they do not completely cover the review content. It is strongly suggested to add further ones, more specific.
Response: We have added more keywords.
1. Introduction
Comments: The Introduction section should be expanded and improved, in order to evidence the aim of the present review. The Authors should better evidence that their review is focused on biocomposites for food packaging applications, as evident form the incipit, and that they reported other potential applications, at the end of the review.
Response: We have included additional sentences into the Introduction and rewritten it so that the main emphasis is focused on the biocomposites for food industry applications.
Comments: The following period “Currently, a novel method to improve the properties of biopolymer films is the use of nanofillers, which can fulfill not only the reinforcing function but also could act as an active ingredient. In recent years, the concept of active agents for biopolymer films has received much more attention. Such active ingredients in biopolymer films can extend the shelf-life of food products, through exhibiting antimicrobial and/or antioxidant activities. The development of nanotechnology has led to the design of nanocomposite film materials in which nanofillers play an active role.” Has to be properly supported with suitable recent literature references, including “Eco-sustainable systems based on poly(lactic acid), diatomite and coffee grounds extract for food packaging, Intern J Biolog Macromolecules 112 (2018): 567-575.” and “Poly(lactic) acid fibers loaded with mesoporous silica for potential applications in the active food packaging, AIP Conference Proceedings 1738 (2016): 270018 (1-4).”
Response: Thank You for this suggestion. The correction has been made.
2. Types of Biopolymers and Nanofillers
Comments: The following statement “This solution applies to the production of cheap, biodegradable packaging material, which is based on biopolymers and nanofillers.” is not totally true, since another drawback of these materials consists in the high cost with respect to the traditional polymers.
Response: The sentence has been corrected.
Comments: The sentence “Blending the different biopolymers together is one of the promising methods of improving the functional properties of biopolymer-based films.” Has to be supported with suitable literature references, including “Electrospun PHBV/PEO co-solution blends: microstructure, thermal and mechanical properties, Materials Science and Engineering: C 33[3] (2013): 1067–1077” and “Tailoring the properties of electrospun PHBV mats: co-solution blending and selective removal of PEO, European Polymer Journal 49 (2013) 3210–3222”.
Response: Thank you for the suggestions. The references have been added.
1.1. Types of biopolymers matrix
Comments: Firstly, please replace 1.1 with 2.1.
Response: The subchapter numbering has been corrected.
Comments: Moreover among the synthetic biopolymers, polycaprolactone has to be included, since it is widely employed. Thus proper references have to be reported, including “Electrospun poly(e-caprolactone)-based composites using synthesized β-tricalcium phosphate, Polymers for advanced technology 22[12] (2011): 1832–1841”.
Response: Polycaprolactone has been included along with the reference.
2.2. Types of nanofillers
Comments: - It is not correct to classify nanoclays as organic fillers. Please correct it.
Response: We have modified the classification. We thank the Reviewer for pointing this out. After reading many publications, we noticed that there are many different classifications of nanofillers. We decided to qualify nanoclays as a separate group.
2.2.1. Organic nanofillers
Comments: The following sentence “The different types of nanocellulose have been incorporated to many biopolymer films [29-31]” has to be corroborated with more recent literature references, including “Effect of silver nanoparticles and cellulose nanocrystals on electrospun poly(lactic) acid mats: morphology, thermal properties and mechanical behaviour, Carbohydrate Polymers 103 (2014): 22– 31.”.
Response: We have added more reference.
Comments: Similarly, the following consideration “Chitosan and chitin nanoparticles obtained respectively from chitosan or chitin have gained attention as nanofillers, due to their attractive surface area, biocompatibility, non-toxicity and film forming ability [32-35].“ has to be supported with more recent literature references, including ”Biodegradable zein film composites reinforced with chitosan nanoparticles and cinnamon essential oil: physical, mechanical, structural and antimicrobial attributes, Colloids and Surfaces B: Biointerfaces 177(2019): 25-32.
Response: The reference has been included.
3. The Effects of Nanofillers on the Functional Properties of Biopolymer-Based Films
Comments: A paragraph about the effect on the gas barrier properties has to be added, being one of the most important requirements in the case of films for food packaging applications.
Response: The paragraph has beed added.
3.2. The effects of nanofillers on the mechanical properties of polymer-based films
Comments: Concerning the effect of CNC and AgNPs, the following paper “Effect of silver nanoparticles and cellulose nanocrystals on electrospun poly(lactic) acid mats: morphology, thermal properties and mechanical behaviour, Carbohydrate Polymers 103 (2014): 22– 31.”. should be described and discussed, evidencing the synergic influence of different nanofillers if used simultaneously.
Response: The reference and discussion about it has been added.
4. The Functional Application of Biopolymer-Based Films with Nanofillers
4.1. Wound dressing
Comments: For the wound dressing applications, it is suggested to discuss the results of the very recent paper “Hydrogen Sulfide-Releasing Fibrous Membranes: Potential Patches for Stimulating Human Stem Cells Proliferation and Viability under Oxidative Stress, International Journal of Molecular Sciences 19(8) (2018): 2368”.
Response: The reference has been added.
4.4. Tissue engineering application
Comments: The following sentence “Hydroxyapatite (HA) nanoparticles promote adhesion, proliferation, and osteogenic differentiation of osteoblast like cells” needs to be supported with suitable literature references, including “Cationic and Anionic substitutions in hydroxyapatite, In: Handbook of Bioceramics and Biocomposites, Springer International Publishing 2016: 145-211.”, “Poly(L-lactic acid)/calcium-deficient nanohydroxyapatite electrospun mats for murine bone marrow stem cell cultures, Journal of Bioactive and Compatible Polymers26[3] (2011): 225-241” and “Tuning multi-/pluri-potent stem cell fate by electrospun poly(L-lactic acid)-calcium-deficient hydroxyapatite nanocomposite mats, Biomacromolecules 13 [5] (2012): 1350-1360”.
Response: All reference as have been added as the Reviewer suggested.
Reviewer 2 Report
In the section 1.1. the classification of biopolymer should be improved. Generally, synthetic biopolymer means the chemical synthetic and fully bio-based polymers like PLA or partially bio-based polymers such as PBS, bio-PET. Moreover, it is much better included bio-PE, bio-PP, bio-polyurethane (PU). On the other hand, natural polymers are much better to classify polysaccharide (neutral: cellulose, hemicellulose, starch, etc., cationic: chitin, chitosan, anionic: alginic acid, hyaluronic acid, etc, bacterial produced: pullulan, carrageenan etc.), proteins (gelatin, keratin, silk, etc), and others (lipid, lignin, natural rubber, urushiol, etc. (although quite specific DNA as structural materials).
Anyway it has better to separate 1) chemical synthetic polymers and 2) naturally produced polymers.
Author Response
Comments: In the section 1.1., the classification of biopolymer should be improved. Generally, synthetic biopolymer means the chemical synthetic and fully bio-based polymers like PLA or partially bio-based polymers such as PBS, bio-PET. Moreover, it is much better included bio-PE, bio-PP, bio-polyurethane (PU). On the other hand, natural polymers are much better to classify polysaccharide (neutral: cellulose, hemicellulose, starch, etc., cationic: chitin, chitosan, anionic: alginic acid, hyaluronic acid, etc, bacterial produced: pullulan, carrageenan etc.), proteins (gelatin, keratin, silk, etc), and others (lipid, lignin, natural rubber, urushiol, etc. (although quite specific DNA as structural materials).
Anyway it has better to separate 1) chemical synthetic polymers and 2) naturally produced polymers.
Response: The classification has been improved.
Reviewer 3 Report
In this manuscript, the authors have given a thorough and comprehensive review on the nanofiller effect on the properties of biopolymer-based nanocomposite films, including the effects on the physical, mechanical and antimicrobial properties. At the meantime, the authors also give a detailed introduction of the various types and components of nanocomposite, the biomedical applications of the materials, and their perspectives on these nanomaterials. The publication of this manuscript will be of great value for other researchers in this field. Thus, I recommend this work to be published in Polymers. But there are some minor issues in the manuscript which need to be addressed.
1. Some concepts mentioned in the manuscript are not accurate.
The first sentence in the abstract: “Nanocomposite films are a group of materials that consist of compounds mainly produced from biodegradable resources (polysaccharides, proteins, lipids and biodegradable synthetic polymers) and from nanofillers.” However, there are lots of work on nanocomposite films based on nondegradable polymers like PMMA, PVA, PVDF, and silicon-based polymers.
Page 3, line 97. “There are two types of nanofillers: organic and inorganic.” But how about organic-inorganic hybrid nanoparticles?
2. Page 5, figure 1, DNA is not included in the natural biopolymers and nanosheet like graphene is not included in the nanofillers. But there are work on DNA and/or graphene-based nanocomposite membranes.
3. Some words need to be changed and improved. On Page 2, “there are many types of biopolymer films that are characterized by both many advantages and disadvantages.”
4. In Figure 1, the bacterial had better be labeled. And some components like enzymes, receptors and cell membrane are not labeled in Figure 2.
Author Response
In this manuscript, the authors have given a thorough and comprehensive review on the nanofiller effect on the properties of biopolymer-based nanocomposite films, including the effects on the physical, mechanical and antimicrobial properties. At the meantime, the authors also give a detailed introduction of the various types and components of nanocomposite, the biomedical applications of the materials, and their perspectives on these nanomaterials. The publication of this manuscript will be of great value for other researchers in this field. Thus, I recommend this work to be published in Polymers. But there are some minor issues in the manuscript which need to be addressed.
1. Some concepts mentioned in the manuscript are not accurate.
Comments: The first sentence in the abstract: “Nanocomposite films are a group of materials that consist of compounds mainly produced from biodegradable resources (polysaccharides, proteins, lipids and biodegradable synthetic polymers) and from nanofillers.” However, there are lots of work on nanocomposite films based on nondegradable polymers like PMMA, PVA, PVDF, and silicon-based polymers.
Response: The senstence have been corrected.
Comments: Page 3, line 97. “There are two types of nanofillers: organic and inorganic.” But how about organic-inorganic hybrid nanoparticles?
Response: Thank you, we have added more information about hybrid nanofillers.
Comments: Page 5, figure 1, DNA is not included in the natural biopolymers and nanosheet like graphene is not included in the nanofillers. But there are works on DNA and/or graphene-based nanocomposite membranes.
Response: We have added more information in the main text and changed the appearance of Figure 1.
Comments: Some words need to be changed and improved. On Page 2, “there are many types of biopolymer films that are characterized by both many advantages and disadvantages.”
Response: The sentence has been corrected.
Comments: In Figure 1, the bacterial had better be labeled. And some components like enzymes, receptors and cell membrane are not labeled in Figure 2.
Response: The corrections have been made.
Round 2
Reviewer 1 Report
The Authors have followed all the Reviewers’ remarks and suggestions and now the review looks very improved and can be accepted in the present version.